# Temperature and Moisture Gradients Drive the Shifts of the Bacterial Microbiomes in 1000-Year-Old Mausoleums

Xin Li [1,*], Xiao'ai Zhou [1], Chen Wu [2], Evangelos Petropoulos [3], Yongjie Yu [2,3,4,*] and Youzhi Feng [4]

1 College of Chemical Engineering, Nanjing Forestry University, Nanjing 210037, China
2 Key Laboratory of Agrometeorology of Jiangsu Province, Nanjing University of Information Science & Technology, Nanjing 210044, China
3 School of Engineering, Newcastle University, Newcastle upon Tyne NE1 7RU, UK
4 State Key Laboratory of Soil and Sustainable Agriculture, Institute of Soil Science, Chinese Academy of Sciences, Nanjing 210008, China
* Correspondence: xli@njfu.edu.cn (X.L.); yjyu@nuist.edu.cn (Y.Y.)

**Abstract:** Cultural relics conservation and prevention from bacterial deterioration are critical for our historical heritage. Thus far, the variations of the ecophysiological features of deteriorating bacterial communities along gradients of temperature and moisture remain unclear. In this study, we used high-throughput sequencing to investigate the changing pattern of bacterial communities on bricks at different positions along two such gradients in the Two Mausoleums of the Southern Tang Dynasty, which have more than 1000 years of history. We found that the tombs were inhabited by a phylogenetically and functionally diverse bacterial microbiomes. Herein, Proteobacteria (34.5%), Cyanobacteria (31.3%), Bacteroidetes (7.8%) and Actinobacteria (7.4%), as well as 'Amino Acid Metabolism (11.2%)' and 'Carbohydrate Metabolism (10.5%)' accounted for the majorities of their compositional and functional profiles related to biodeterioration. Non-metric scaling in combination with PERMANOVA tests indicated that shifts in bacterial community compositions were governed by temperature, followed by moisture. In addition, we found that tourism-related anthropogenic activities could have played non-negligible roles in community assembly, especially in the areas that account as attractions (i.e., back room of the Qinling Mausoleum). Collectively, this study advances the knowledge regarding the deteriorating microbiomes of cultural monuments, which is essential for the conservation of historical cultural relics.

**Keywords:** cultural relics; biodeterioration; microbial community

## 1. Introduction

In the field of cultural heritage, the conservation of subterranean tombs is of great archaeological importance and plays a vital role in understanding the evolution of our customs and culture. However, complex and diverse microbial communities do colonize such relics, especially when such sites of rich cultural heritage are exposed to the external environment [1–3]. In this regard, microbial colonies can result in adverse physical, chemical, structural and aesthetic alterations damaging the ancient relics and artworks through a process known as biodegradation of cultural heritage [4–6]. Biodeterioration refers to the microbial-related chemical or physical alteration, or even change in appearance, of materials and objects [7]. Archaea, bacteria, algae, fungi, animals and plants often cause biodeterioration of valuable cultural heritages [2,8,9]. At present, conservation of cultural heritages has gained attention; hence, biodeterioration derived from microorganisms on historical relics has become a 'hot' topic.

Environmental factors, including solar radiation, humidity, temperature fluctuations, nutritional characteristics and/or anthropogenic activities, are generally considered to be related to microbe-driven biodeterioration [6,10]. The microbial community that colonizes on/in the cultural heritage is one of the main causes exhibiting obvious spatial patterns

depending on site conditions [11–13]. For example, previous studies have shown that consecutive freeze–thaw cycles can notably change the internal structure of cultural heritages and provide shelter for microbial growth, resulting in an accelerated biodeterioration of stone cultural relics [5,14]. Exposure to sunlight benefits phototrophic microorganisms, such as algae and cyanobacteria. A large number of bacteria and fungi will also thrive on the surface of murals due to high temperature and humidity [6,15–17]; this will inevitably erode the cultural heritages and reduce their aesthetic value [18]. The microbial outbreak in the Lascaux Cave is strong evidence that environmental factors govern the microbial community in the historical relics [19].

Among all environmental factors, indoor temperature and moisture conditions impose a pivotal influence on microbial communities colonizing cultural relics [20–24]. One recent investigation has found that changing patterns of bacterial communities align well with temperature and moisture gradients in eight karst caves in southwest China [25]. However, how temperature and moisture conditions govern compositional and functional features of deteriorating microbiomes on cultural relics is still elusive. In addition, besides biotic factors, anthropogenic activities also can influence the microbial community composition on relics [26]. For instance, the microbial outbreak in the Lascaux Cave is the best case, due to the tourism [19]. In this respect, we hypothesized that in subterranean historical cultural sites, hydrothermal conditions control the deteriorating microbial community, and in some specific positions the influence of anthropogenic activity cannot be neglected. Such information would extend our understanding of the conservation of cultural heritage.

For this purpose, we investigated the changing patterns of bacterial communities in the two mausoleums of the Southern Tang Dynasty, an imperial cemetery in eastern China and one of the largest imperial mausoleums since the Southern Tang Dynasty (more than 1000 years ago). Two similar tombs, i.e., the Shunling Mausoleum and Qinling Mausoleum in the imperial cemetery, have been open to the public for more than 70 years. A huge amount of crucial archaeological information and material has been discovered since 1950. There are obvious gradients of temperature and moisture in the two mausoleums [27]. These data provide an ideal platform to unravel how the microbial community changes along different temperature and moisture gradients in a subterranean ecosystem. Furthermore, the microbial community structure determines its ecological function, including the individuals within the colony participating in the biodegradation of organic compounds through acid production and enzyme production, processes that damage cultural relics [20,21,28]. In this study, high-throughput sequencing-based analyses were used to determine the bacterial community compositions, and PICRUSt was applied to predict their ecological functions along different hydrothermal gradients in the two mausoleums of the Southern Tang Dynasty. Therefore, this study firstly tried to clarify the taxonomic composition and the spatial distribution pattern of these two tombs; secondly, the study tried to understand how temperature and moisture gradients govern spatial succession patterns in bacterial communities.

## 2. Materials and Methods

### 2.1. Sampling and Environmental Monitoring

This study was conducted in a 1000-year-old mausoleum located at the southern foot of Zutang Mountain in Nanjing, China (31°76–32°87′ N, 117°50–118°90′ E). This ancient mausoleum was excavated and named Two Mausoleums of the Southern Tang Dynasty by Nanjing Museum in 1950. There are two separate tombs in this ancient mausoleum, i.e., Shunling Mausoleum and Qinling Mausoleum. The Shunling Mausoleum is 21.9 m long and 10.12 m wide; the Qinling Mausoleum is 21.48 m long and 10.45 m wide. Both mausoleums include the entrance, the front room, the middle room and the back room. Particularly, the Qinling Mausoleum is by far the largest, most complex and most decorated imperial mausoleum in southeast China and is well known for its exquisite murals.

In September 2020, samples were collected from the bricks of the following chambers: the entrance, front room, middle room and back room of the Shunling Mausoleum and



Qinling Mausoleum. Eight samples were taken at each position—a total of 64 samples. Microbial colonies from the brick surface were carefully scraped using a sterile scalpel. A sketch map of the Shunling Mausoleum and Qinling Mausoleum is shown in Figure 1. All samples were immediately placed on ice and rapidly returned to the laboratory where they were stored at 4 °C for further analysis.

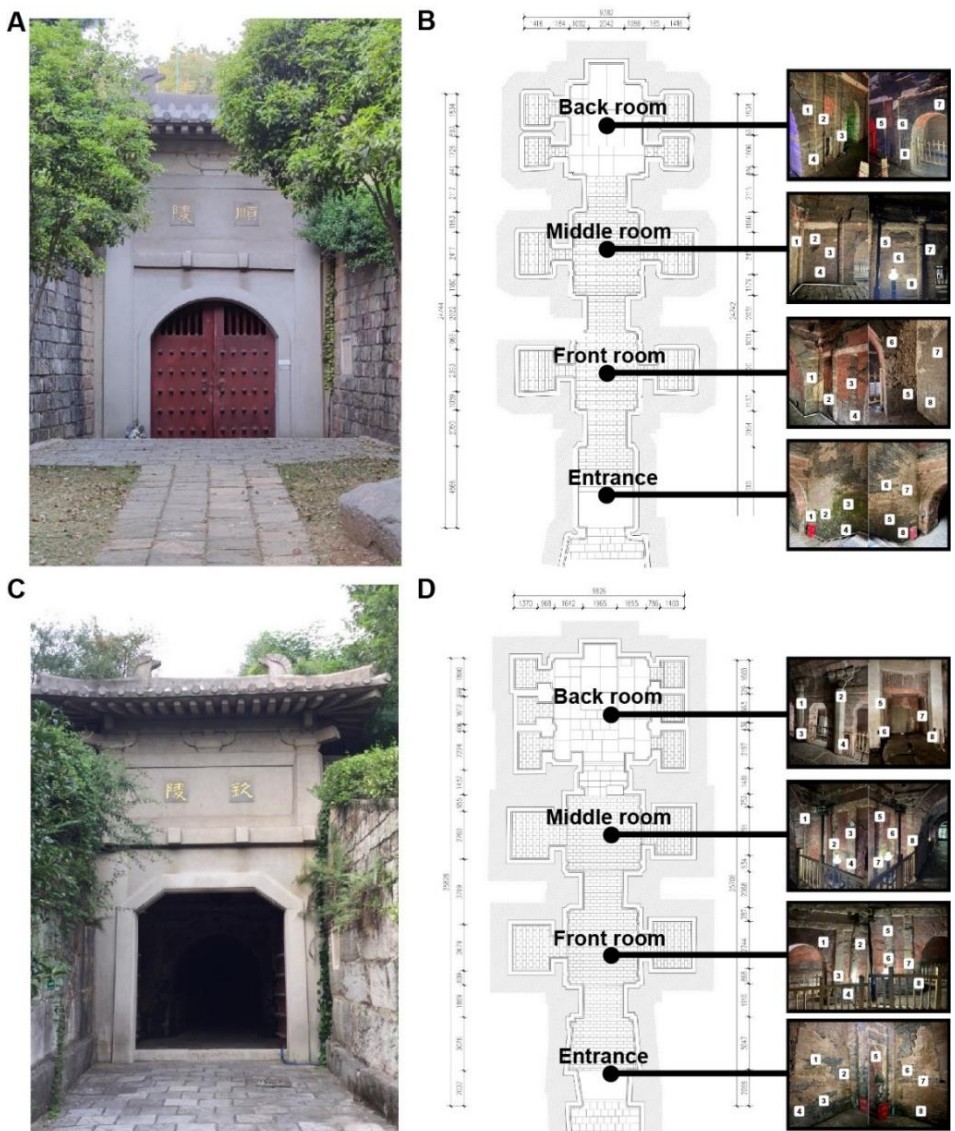

**Figure 1.** Shunling Mausoleum (**A**) and Qinling Mausoleum (**C**) have a 1000 years history and have been accessible for over 70 years. The sampling positions in the Shunling Mausoleum (**B**) and Qinling Mausoleum (**D**). The numbers in the figure (**B**,**D**) represented sampling sites in different rooms.

The Two Mausoleums of the Southern Tang Dynasty have been open to the public for over 70 years, and the entrances have been exposed to sunlight for a relatively long time. The temperature and moisture inside the mausoleums were measured by electronic temperature and moisture content recorders (HOBO; MX2301A); monitoring lasted from 25 April 2020 to 6 November 2021, and logging took place at 30 min intervals.

## 2.2. DNA Isolation, PCR Conditions

DNA was extracted from 0.5 g of each sample using the FastDNA SPIN Kit for soil (MP Biomedicals, Santa Ana, CA, USA) following the manufacturer's instructions. The extracted DNA was eluted in 50 µL of TE buffer and then quality analyzed by gel electrophoresis. Af-

ter that, the extracted DNA concentration was quantified using the NanoDrop-2000 (Themo Fisher Scientific, Waltham, MA, USA) and stored at $-20$ °C for further analysis. PCR amplification was conducted using the 519F (5′-CAGCMGCCGCGGTAATWC-3′)/907R (5′-CCGTCAATTCMT TTRAGTTT-3′) primer set targeting the V4–V5 region of the bacterial 16S rRNA gene. The 5 bp barcoded oligonucleotide sequences were fused to forward primers to distinguish between different samples. Amplifications were carried out using the following conditions: 94 °C for 5 min, 30 cycles of 94 °C for 30 s, 55 °C for 30 s, and 72 °C for 45 s, and a final extension at 72 °C for 10 min. Reaction products for each sample were pooled and purified using the QIAquick PCR Purification Kit (Qiagen, Hilden, Germany) and quantified using NanoDrop-2000 (Thermo Fisher Scientific, Waltham, MA).

*2.3. Processing of High-Throughput Sequencing Data*

The PCR products from all the samples were sequenced with Illumina Miseq sequencing platform (Illumina Inc.). The barcoded PCR products from all samples were normalized in equimolar amounts prior to sequencing. The 16S rRNA gene data were analyzed using the Quantitative Insights into Microbial Ecology (QIIME) pipeline for data sets (http://qiime.source.org) (10 November 2020) [29]. The sequences with either one mismatched base in the barcode or a quality score <25 and a sequence length less than 200 bp were removed, and the rest of the reads were assigned to samples based on barcodes. After denoising and chimera filtering using URAPRSE, the valid reads were then binned into operational taxonomic units (OTUs) based on 97% sequence similarity. The most abundant sequence from each OTU was selected as the representative sequence. Taxonomy was assigned to bacterial OTUs with reference to a subset of the SILVA 138 database (http://www.arb-silva.de/download/archive/qiime/) (accessed on 15 November 2020). In total, 3,864,355 reads of bacterial 16S rRNA gene fragments in Shunling Mausoleum and 3,642,092 reads of high-quality bacterial 16S rRNA in Qinling Mausoleum were obtained for downstream ecological analyses. For the OTU-based analyses, the original OTUs were rarefied to a depth of 50,000 reads per sample in Shunling and Qinling Mausoleums to measure both diversity and species-composition patterns.

*2.4. Statistical Analysis*

Species-level based Observed Taxonomical Units (OTUs) were used to characterize the community's alpha diversity. Non-metric multidimensional scaling (NMDS) was performed to visualize the dissimilarities of bacterial community composition between different positions according to the Bray-Curtis (OTUs level) distances. Permutational multivariate analysis of variance (PERMANOVA) [30] was conducted to test for statistically significant differences in community composition. The above analysis was performed using the "*Vegan*" package (version package, Version 4.0.4) in R software. Analysis of variance (ANOVA) was used to analyze the differences in the diversity index between groups, and SPSS16.0 was used to determine the difference. Spearman correlations between the Observed OTUs and tombs' environmental factors were calculated using the "*corr.test*" function in the "*psych*" package in R [31]. Mantel tests were conducted between environmental factors (temperature and/or moisture content) and bacterial community using R software. Linear regressions between Bray–Curtis distance and changes in temperature and moisture content were investigated to determine the relationship between bacterial communities and different environmental conditions. For all the statistical tests, significance and high significance were determined as $p < 0.05$ and $p < 0.01$, respectively.

Functional characteristics of the bacterial community under the hydrothermal gradient were analyzed by Phylogenetic Investigation of Communities by Reconstruction of Unobserved States (PICRUSt) [32]. In brief, the normalized OTU table was submitted to the local PICRUSt database (version 1.1.2) for genome prediction, followed by creation of final genomic functional predictions based on Kyoto Encyclopedia of Genes and Genomes (KEGG) pathways. The functional differences of bacterial communities in different locations were compared with heatmap analysis.

## 3. Results

### 3.1. The Hydrothermal Conditions in the Mausoleums

An obvious temperature and moisture gradient was found at different positions in Shunling and Qinling Mausoleums after several months of monitoring (from 25 April 2020 to 6 November 2020) (Figure 2). The highest temperature and the lowest moisture content were observed at the entrance. On the contrary, the lowest temperature and highest moisture content were observed in the back room.

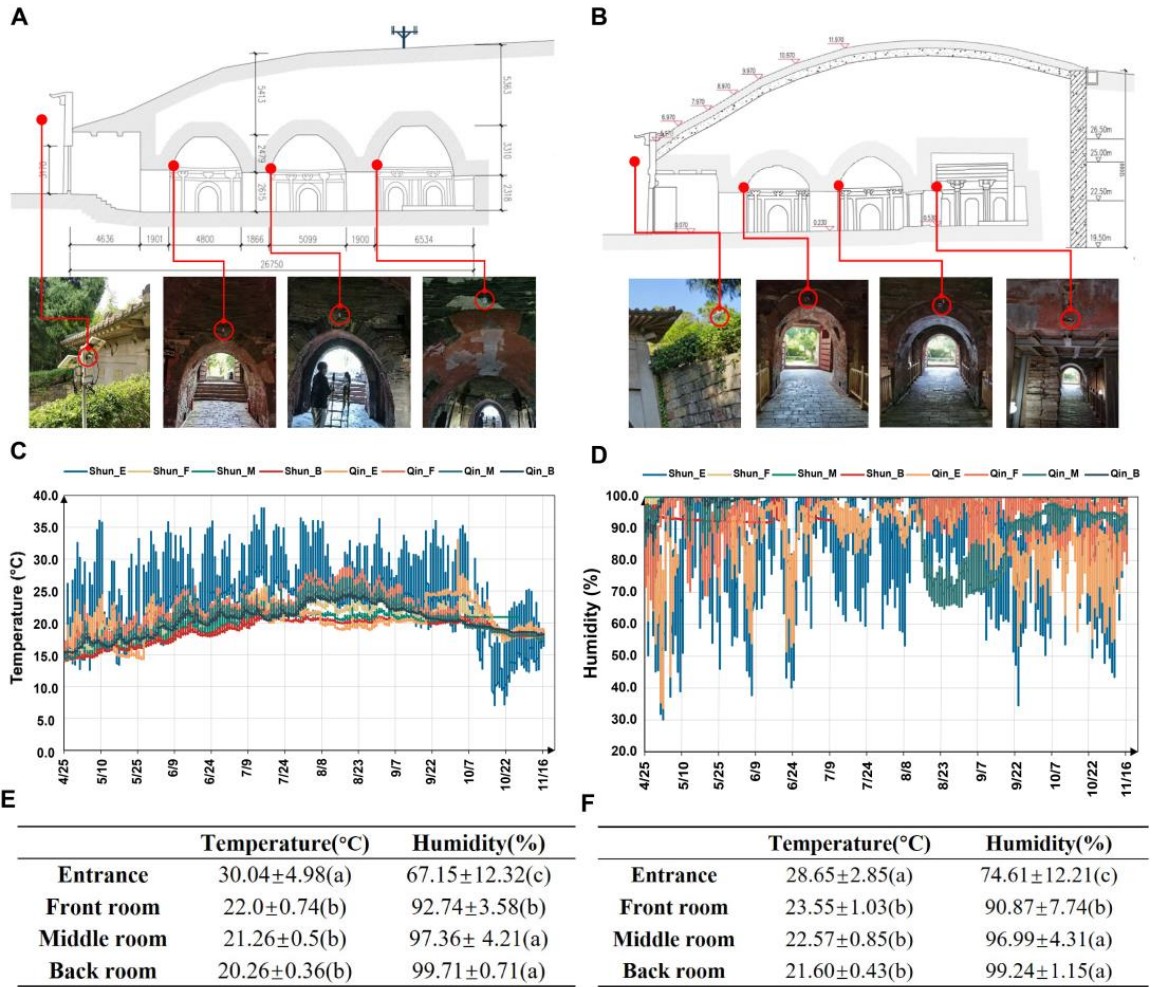

**Figure 2.** Monitoring data of temperature and humidity in the Shunling and Qinling Mausoleums. Electronic recorders were installed at different positions in Shunling (**A**) and Qinling Mausoleums (**B**). The temperature and humidity in Shunling (**C**) and Qinling Mausoleums (**D**) from 25 April to 6 November 2020. The temperature and humidity in Shunling (**E**) and Qinling Mausoleums (**F**) at sampling time. The values are the means of eight replicates (mean ± SD). The lowercase numbers (**A**,**B**) indicated the dimensions of Shunling and Qinling Mausoleums. Values followed by different lowercase letters (**E**,**F**) differ significantly ($p < 0.05$). Abbreviations: Shun_E, Shunling entrance; Shun_F, Shunling front room; Shun_M, Shunling middle room; Shun_B: Shunling back room; Qin_E, Qinling entrance; Qin_F, Qinling front room; Qin_M, Qinling middle room; Qin_B, Qinling back room.

### 3.2. Bacterial Community Shifts and Their Driving Environmental Factors

Based on the results of OTU species annotation, the dominant bacteria with the highest relative abundance were selected at the phylum level to demonstrate the differences of bacterial species between the different positions in mausoleums. A total of seven major bacterial phyla were identified in the two tombs: Proteobacteria, Cyanobacteria, Bacteroidetes,

Acidobacteria, Actinobacteria, Planctomycetes and Firmicutes (Figure S1). Sequences affiliated with Proteobacteria, Cyanobacteria and Bacteroidetes dominated in Shunling Mausoleum, ranging from 34.3% to 41.4%, from 15.6% to 42.6% and from 7.3% to 15.4%, respectively. In Qinling Mausoleum, Cyanobacteria, Proteobacteria and Actinobacteria were dominant, with 29%~37.6%, 21.2%~34.7% and 5.2%~20.3%, respectively. The bacterial taxa were similar in each mausoleum, but the relative abundance of each phylum varied between the two mausoleums (Figure S1).

Species-level based observed OTUs (namely richness) were used to characterize community alpha diversity (Figure 3). In the Shunling Mausoleum, the alpha diversity at the entrance was higher than that of other sampling positions (the front, middle and back rooms) ($p < 0.05$). The entrance of Qinling Mausoleum showed a significantly higher alpha diversity than the front and middle rooms ($p < 0.05$). Interestingly, a different bacterial community diversity was observed in the back room of Qinling Mausoleum. The highest alpha diversity among all the studied sites in the Qinling Mausoleum was found in the back room (Figure 3). No significant differences in bacterial richness were observed between the two entrance sites of the Shunling and Qinling Mausoleums ($p > 0.05$).

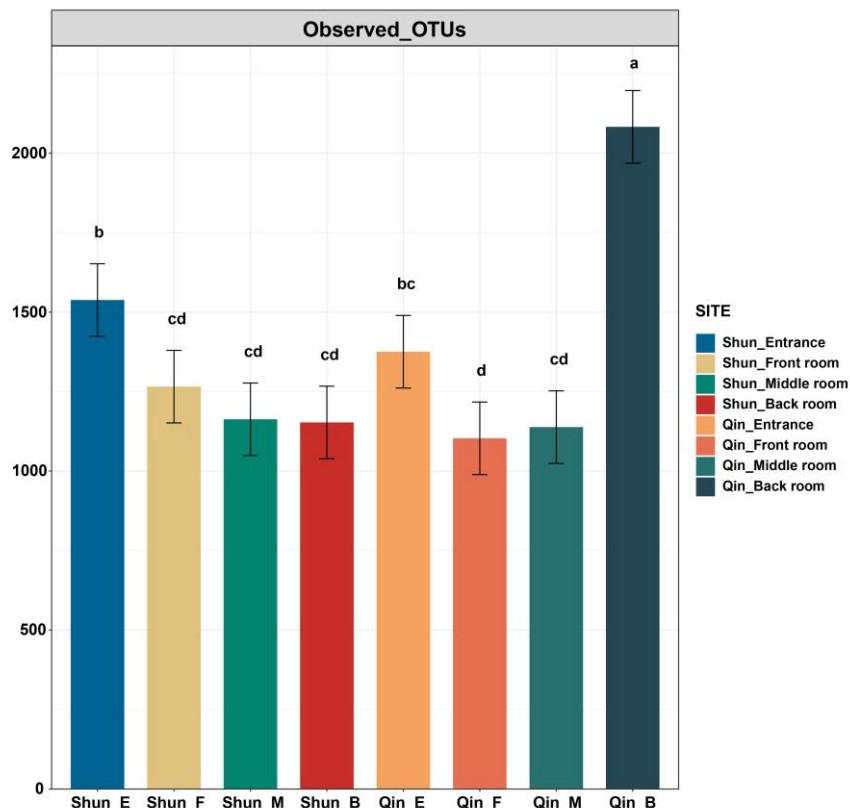

**Figure 3.** OTU richness among different positions of Shunling and Qinling Mausoleums. Different lowercase letters over error bars denote significant differences ($p < 0.05$). Abbreviations: Shun_E, Shunling entrance; Shun_F, Shunling front room; Shun_M, Shunling middle room; Shun_B: Shunling back room; Qin_E, Qinling entrance; Qin_F, Qinling front room; Qin_M, Qinling middle room; Qin_B, Qinling back room.

### 3.3. Diversity of Bacterial Community

The bacterial community of all sample sites was significantly clustered into different groups in both studies of the ancient mausoleums ($p < 0.05$) (Figure 4). In Shunling Mausoleum, the heterogeneity of the bacterial communities in the middle room was the highest, indicating that the bacterial community composition in the middle room was the most complex (Figure 4A). However, in Qinling Mausoleum the bacterial community in the back room was the most heterogeneous, and its composition was the most complex. The

results of PERMANOVA analysis showed that there were significant differences in bacterial community structure between four positions in the Shunling or Qinling Mausoleums ($p < 0.05$) (Tables S1 and S2).

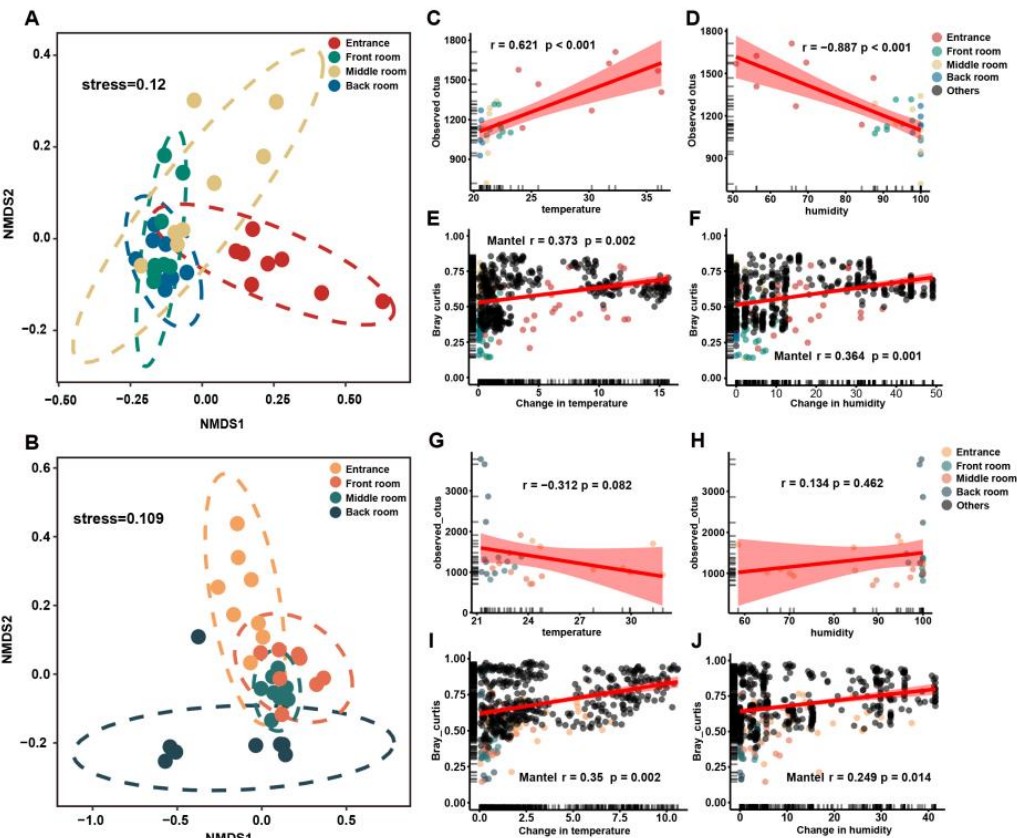

**Figure 4.** Nonmetric multidimensional scaling (NMDS) analysis of bacteria in Shunling (**A**) and Qinling Mausoleums (**B**) based on Bray–Curtis distance. Spearman Correlation analysis of observed OTUs with temperature and humidity in Shunling (**C,D**) and Qinling Mausoleums (**G,H**). Distance matrix regressions between Bray–Curtis and hydrothermal gradient in Shunling (**E,F**) and Qinling Mausoleums (**I,J**). Horizontal axes indicate Euclidean distances based on temperature and humidity. Linear relationships were evaluated by Mantel test (*p*-value provided for each panel) (**E,F,I,J**). Linear models (shown as red lines) and associated correlation coefficients are provided for each panel.

In addition, the statistical analyses found significant relationships (Spearman correlations) between species richness in the Shunling Mausoleum and environmental factors (temperature and moisture content). In Shunling Mausoleum, temperature (Figure 4A) was significantly positively correlated with species richness ($p < 0.001$), while moisture content (Figure 4B) was significantly negatively correlated with observed OTUs ($p < 0.001$). However, the correlation between species richness and temperature (Figure 4E) and humidity (Figure 4F) was not significant in the case of the Qinling Mausoleum ($p > 0.05$). Mantel tests between environment factors and bacterial community compositions of the Shunling Mausoleum and Qinling Mausoleum indicated that temperature ($r = 0.373$, $p = 0.002$ and $r = 0.35$, $p = 0.002$) and humidity ($r = 0.364$, $p = 0.001$ and $r = 0.249$, $p = 0.014$) influenced bacterial community compositions. Distance matrix regressions revealed a significant positive association between the bacterial communities and the changes in temperature and moisture content in the Shunling Mausoleum (Figure 4C,D); the correlation between bacterial communities and changes in temperature ($r = 0.373$, $p = 0.002$) was higher than that of changes in humidity ($r = 0.364$, $p = 0.001$). A similar pattern was also found between the bacterial community on the brick and changes in temperature and humidity for the Qinling Mausoleum (Figure 4G,H).

The observed OTUs in the back room of the Qinling Mausoleum were significantly higher than those in any other area of the tomb (Figure 1). Consequently, using Venn plots to define the endemic OTUs' taxa in the back room of the Qinling Mausoleum, it was revealed that 459 OTUs are common to the four positions of the Qinling Mausoleum, while 88 OTUs from the front room are unique; 120, 140, and 1142 OTUs were unique to the entrance, the middle room and the back room, respectively (Figure 5A). The 1142 unique OTUs from the back room were clustered at the family level (Figure 5B). The top species in the back room were mainly affiliated with *Gaiellaceae*, *Anaerolineaceae*, *Acetobacteraceae*, *Ruminococcaceae*, *Streptosporangiaceae*, *Sphaerobacteraceae* and *Alicyclobacillaceae*, which were unique bacterial taxa compared to the outside soils (Figure S2).

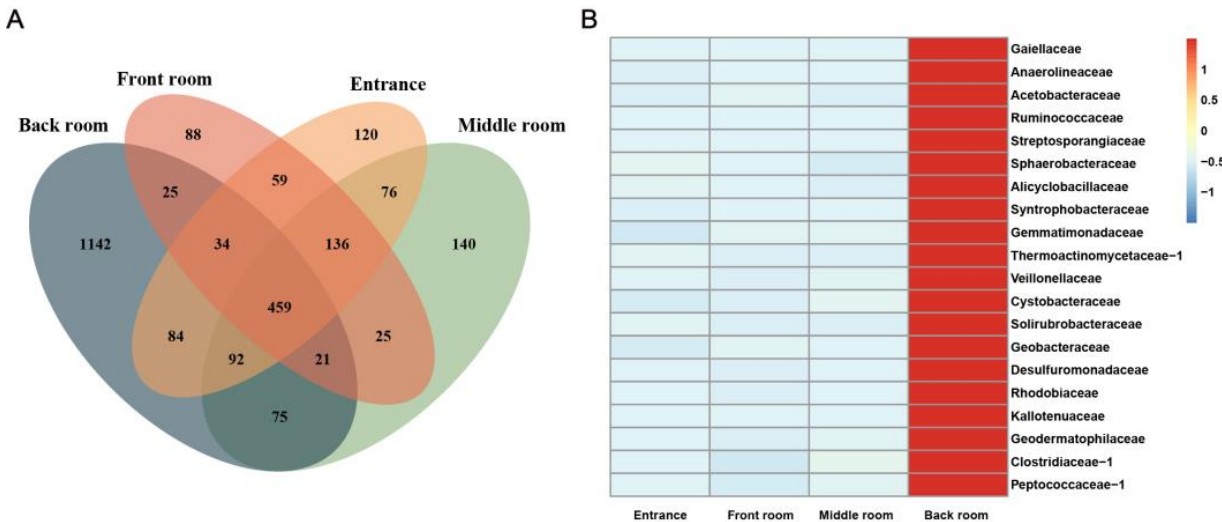

**Figure 5.** Endemic OTUs' taxa in the back room of Qinling Mausoleum. (**A**) Venn plot of OTUs' taxa among four positions in Qinling Mausoleum. (**B**) Heat map of family level species annotation of endemic OTUs for the top 20 abundances in Qinling Mausoleum back room.

### 3.4. PICRUSt for Function Prediction

The bacterial metabolic activity on the bricks from all areas (the entrance, front room, middle room and back room) are remarkably different within the Shunling Mausoleum (Figure 6A). 'Amino Acid Metabolism', 'Environmental Adaptation' and 'Xenobiotics Biodegradation and Metabolism' occupied a notably higher proportion in the entrance, while 'Biosynthesis of Other Secondary Metabolites', 'Energy Metabolism' and 'Metabolism of Cofactors and Vitamins' occupied a higher proportion in the back room. The abundance of specific species within the bacterial community reveals that the functions of the front and middle rooms were generally low. Regarding carbon metabolism, the bacteria from the entrance area could maintain the following potential degradation functions: 'Amino Acid Metabolism', 'Lipid Metabolism' and 'Biodegradation and Metabolism'. In addition, the bacteria from the back room had a higher relative abundance of species related to 'Metabolism of Cofactors and Vitamins' and 'Metabolism of Terpenoids and Polyketides'. However, the functional abundance of bacterial community in the middle room was notably higher than that of any other position in the Qinling Mausoleum (Figure 6B).

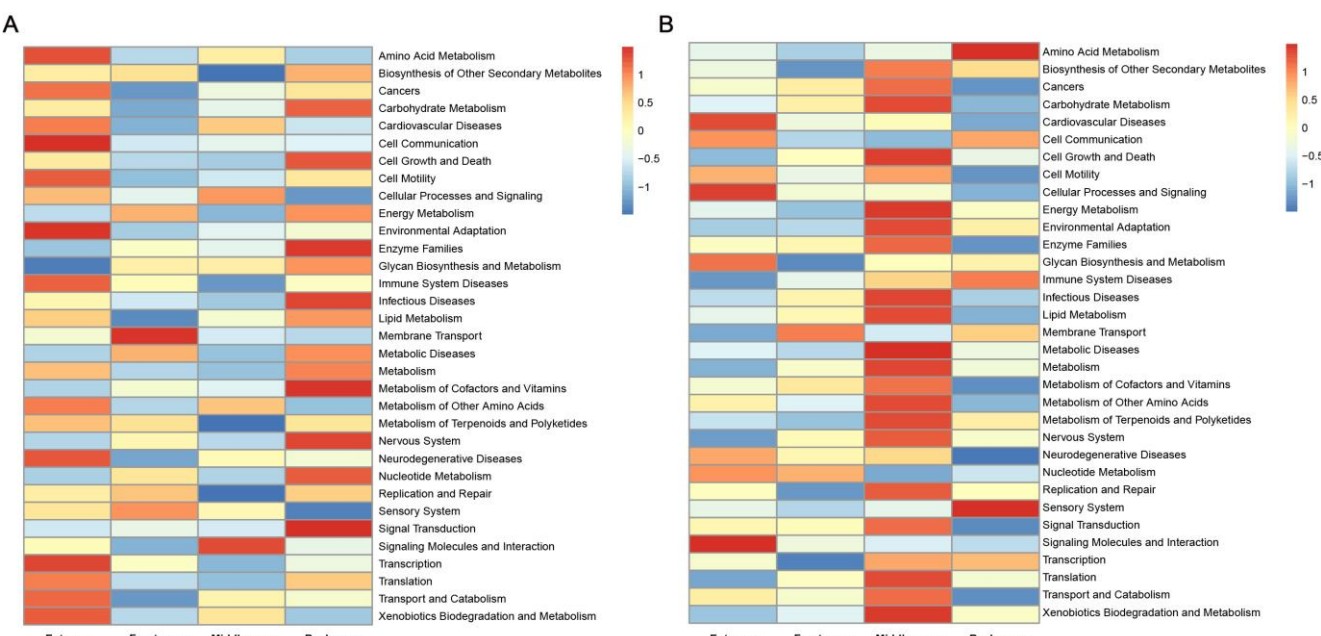

**Figure 6.** Heat map of the KEGG level-2 functional pathways for communities on bricks among the entrance, front, middle and back rooms in Shunling (**A**) and Qinling Mausoleums (**B**). The normalized relative abundance of each KEGG pathway is indicated by a gradient of color from blue (low abundance) to red (high abundance).

## 4. Discussion

### 4.1. Changing Patterns of Microbial Community along the Temperature and Moisture Gradients

Temperature and humidity are the two main environmental factors that drive the microbial distribution in natural ecosystems [33–36]. Once ancient cultural relics were open to the public, the microorganisms would be ecologically distributed and changed according to the environmental conditions, especially according to the temperature and moisture conditions [17,37,38]. For example, K. Frindte et al. [39] found that temperature and soil moisture control microbial community composition in microtopographic ecosystems. Ma et al. [21] also reported that temperature could influence the bacterial communities in a cave environment. In this study, the bacterial diversity was higher at the entrance than in the front and middle rooms in Shunling Mausoleum. The temperature of the entrance was found to be higher than that of the front and middle rooms in Shunling Mausoleum; the moisture of the entrance was lower than that of other sites. These phenomena are in line with the hypothesis that temperature and moisture conditions control the distribution of microbes in the ancient cultural relics.

However, the temperature and moisture conditions had different influences on bacterial patterns [36,40]. In this study, temperature decreased from the entrance to the back room, while the moisture increased from the entrance to the back room in both mausoleums. Specifically, the decrease of temperature decreased Proteobacteria but increased Actinobacteria, while the decrease of moisture increased Proteobacteria but decreased Actinobacteria. For Proteobacteria, it has been demonstrated that this phylum is advanced in the ability to degrade a wide range of organic compounds [20,41]. A more abundant organic compound can be provided at proper temperature and moisture conditions. It was found that 'Amino Acid Metabolism', 'Environmental Adaptation' and 'Xenobiotics Biodegradation and Metabolism' occupied a notably higher proportion at the entrance. As a result, Proteobacteria dominated the microbial community at the entrance sites, which hold higher temperature and lower moisture conditions than that of the back rooms of the mausoleums. Similar trends of Proteobacteria have also been reported for other tombs or caves, such as Altamira caves, Heshang Cave and the tomb of Emperor Yang of the Sui

Dynasty [20,21,42]. For Actinobacteria, it often dominant inside of cultural relics, such as Dahuting Han Dynasty Tomb, the Altamira caves, and the Roman catacombs [1,28,43]. Actinobacteria can drive biodeterioration via the production of organic acids, the formation of biofilms, the penetration of relic materials and/or the excretion of a mixture of cellulolytic enzymes. Therefore, the low temperature and high moisture conditions in the back rooms of the mausoleums are more suitable for the survival of Actinobacteria.

### 4.2. Tourism-Related Activity Drives Microbial Community Shift

The bacterial diversity in the back room of the Shunling Mausoleum was significantly lower than that of the back room of the Qinling Mausoleum, although the temperature and moisture conditions are very similar between the two back rooms of Shunling and Qinling Mausoleums. The only reason why the bacterial diversity between the two studied mausoleums differed might be the different tourism-related activities. The Qinling Mausoleum is the first royal tomb with exquisite frescoes discovered in southeast China. The exquisite frescoes are located in the back room of the Qinling Mausoleum, which attracted larger numbers of visitors to the back room due to interest in the frescoes. The difference may have been caused by the tourist activity [13,44,45]. Therefore, it can be assumed that the tourism-related activity could have promoted the shifts in the microbial community diversity in the back room of the Qinling Mausoleum.

The influences of tourism-related activity on microbial community may consist of four aspects: external microbes, external substances, artificial light and elevated $CO_2$. Firstly, the unique species in the back room of the Qinling Mausoleum mainly belong to *Gaiellaceae*, *Anaerolineaceae*, *Rhodospirillaceae*, *Planctomycetaceae Ruminococcaceae*, etc. Interestingly, *Anaerolineaceae* [46], *Ruminococcaceae*, *Clostridiaceae* and *Peptococcaceae* can all be found in the human oral microbiome database (www.homd.org) (accessed on 11 December 2022). These observations provide a sound clue to support that tourism-related activity brought external microbes to the back room of Qinling Mausoleum. Secondly, human visitors bring a variety of external substances to subterranean sites, such as hair, secreted sweat, clothing fibers and dust that could potentially provide nutrients for microorganisms [47]. In this study, *Veillonellaceae* was found in the back room of the Qinling Mausoleum. Previous study has been demonstrated that *Veillonellaceae* can play an important role in the development of oral biofilms and human oral ecology [48]. Therefore, humans can directly affect the air quality inside tombs by breathing, coughing and sneezing. Related droplets contain a large number of microorganisms originating from the oral mucosa, which could be a source of bacteria and fungi in the air of tombs [49]. Thirdly, intense artificial lighting for visitors could lead to the uncontrolled development of photosynthetic microorganisms, such as Cyanobacteria and algae [50–53]. In this study, a significant increase of Cyanobacteria was found in the back room of the Qinling Mausoleum. It should be noted that the colonized Cyanobacteria could cause aesthetic, physical and chemical damages to historical relics [19,44,51]. Fourthly, inspiration from numerous visitors has raised the $CO_2$ concentration levels inside the tomb [44,54], intensifying the corrosion of the walls and facilitating microbial proliferation that could accelerate biodeterioration [49,55]. Therefore, although the development and involvement of cultural relics in the tourism industry is beneficial for the local economy, tourism-related activity could bring additional 'non-native' microbial resources that may inevitably damage the 'skins' of the ancient relics due to their uniqueness in the area's metabolic habits.

### 5. Conclusions

In this study, we found that temperature and moisture gradients drive the bacterial community shifts in Two Mausoleums of the Southern Tang Dynasty, which have a history of over 1000 years. The composition of the bacterial microbiome was mainly controlled by temperature and moisture content. In addition, tourism-related anthropogenic activities could have resulted in a perturbation of bacterial community composition in such environments. Such environmental gradients mainly contribute to the diversity of bacterial

communities. Our findings give a better understanding of the subterranean biodiversity in such partially enclosed engineered eco-systems, providing insight about the preservation of subterranean relics.

**Supplementary Materials:** The following supporting information can be downloaded at: https://www.mdpi.com/article/10.3390/atmos14010014/s1, Table S1: Dissimilarity tests of bacterial communities among different positions in the Shunling Mausoleum using Permutational multivariate analysis. Table S2: Dissimilarity tests of bacterial communities among different sites in the Qinling Mausoleum using Permutational multivariate analysis. Figure S1: Relative abundances of bacterial community composition at phylum level on the bricks in the Shunling (A) and Qinling mausoleums (B). Figure S2: Endemic OTUs taxa in the back room of Qinling Mausoleum. (A) Venn plot of OTUs taxa between soil outside the mausoleum and the back room of Qinling Mausoleum. (B) Heat map of family level species annotation of endemic OTUs for the top 20 abundances in Qinling Mausoleum back room.

**Author Contributions:** Conceptualization, X.L., Y.Y. and Y.F.; methodology, X.L., X.Z. and C.W.; formal analysis and investigation, X.L., X.Z., E.P. and Y.Y.; writing—original draft preparation, X.L., X.Z. and C.W.; writing—review and editing, E.P., Y.Y. and Y.F.; funding acquisition, Y.Y. and Y.F. All authors have read and agreed to the published version of the manuscript.

**Funding:** This research was funded by National Key R&D Program, grant number 2019YFC1520700.

**Institutional Review Board Statement:** Not applicable.

**Informed Consent Statement:** Not applicable.

**Data Availability Statement:** The amplicon sequences reported in this article have been deposited in the NCBI BioProject (accession nos. PRJNA913085). All the other study data are included in the article and/or supporting information.

**Conflicts of Interest:** The authors declare no conflict of interest.

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
