# Peer review of "Temperature and Moisture Gradients Drive the Shifts of the Bacterial Microbiomes in 1000-Year-Old Mausoleums"

_atmosphere, doi:10.3390/atmos14010014_

Round 1

Reviewer 1 Report

The microbe-driven biodeterioration is critical for the conservation and prevention of ancient cultural relics. This manuscript performed an interesting study that linked the microbiome and different temperature and moisture contents, with the appropriate contrasting plots and numbers of replication in the ancient mausoleums, such studies are scarce and warrants merit. Though the soil bacterial community cannot reflect all the biodeterioration of cultural relics, the results in this study can clearly reveal the significant differences in microbial communities between different positions in the investigated ancient mausoleums. The further statistical analysis in this manuscript demonstrated the distribution of bacterial community associated with the temperature and moisture gradients and tourism-related activities. The data and interpretation in this study are reasonable and reliable. It is a sound and solid study, which could give a heuristic and pragmatic value for the conservation and prevention of cultural relics in the future. In my opinion, it can be accepted for publication after minor modifications. 

Minor concerns:

1. Abstract section: line 23, ‘PERMANOVA tests’

2. Line 101: The microbe-driven biodeterioration occurs continuously all the time. Why was the sampling time selected in September in this study?

3. Line 118: Please add a space between ‘0.5’ and ‘g’. Please check the whole paper and add a space between the number and unit.

4. Line 123: The primer set 519F/907R is only targeting bacteria. What are the responses of fungal community among different sample sites?

5. Line 143: The sequencing depth is not the same between the two mausoleums. Please give a further explanation on the sequencing depth.

6. Line 156: omit ‘[25]’. Please check the references in the whole paper.

7. Lines 174 and 175: Omit ‘area’.

8. Line 346: The influence of light is not investigated in this manuscript.

9. Line 347: The letter of ‘t’ in ‘two’ should be written in capital.

Author Response

On behalf of all authors, I would like to express our sincere appreciation to you for the time and the efforts put into our manuscript. We are confident that with their illuminating comments and their significant suggestions our work has significantly improved. We would also like to thank the chief editor for his/her perspicacious editorial work.  The point-by-point responses have been made. Please find it in the WORD file. 

Reviewer 2 Report

The manuscript “Temperature and moisture gradients drive the shifts of the bacterial microbiomes in 1000-year-old mausoleums” deals with the change of bacterial communities along gradients of temperature and moisture in the Two Mausoleums of the Southern Tang Dynasty. The article also attempts to clarify the relationship between tourism-related anthropogenic activities and microbial composition. I believe that this manuscript is interesting, has quality and is relevant to the area, thus being worthy of publication after revisions below are conducted.

1. Line 19-20, 289-291, 296-298, 304-305 mentioned the dominant phylum identified in the Two Mausoleums of the Southern Tang Dynasty, but no detailed results were given in the manuscript. Please described the dominant bacterial community in the Results part.

2. Figure 2C and 2D, please revise the letters representing significance in Moisture and Humidity, for example, use only a, b, c.

3. Line 187-188 mentioned a clear trend of decreasing alpha diversity from the entrance to the back room, but the Figure 3 showed no significant difference in alpha diversity among the front, middle and back houses. Please revise or check the description.

4. Line 216-220. The r value was below 0.4, which represents a weak correlation between environmental factors and the bacterial community. However, the authors stated that “temperature and humidity strongly influenced bacterial community compositions”. Please revise such description, and make sure it qualified the statistical analysis.

5. Line 274-277, “Once an ancient cultural relic was open to the public, the microorganisms would be ecologically distributed occurred according to the environmental conditions, especially according to the temperature and moisture conditions”. The ecological distribution of microorganisms should be “changed” rather than “occurred” with the opening of the cultural relics to the public.

6. Line 280-281, based on Figure 3, in Qinling mausoleums, there was no significance between the bacterial diversity at the entrance site and in the middle room. Please revised the sentence in the manuscript.

7. Line 282-283 “the moisture was the entrance found lower than that of other sites” should be converted to “the moisture of the entrance was lower than that of other sites”? Please check the structure of this sentence.

8. Line 322-323, Gaiellaceae, Anaerolineaceae, Ruminococcaceae, Clostridiaceae and Peptococcaceae are also common in soil, what made you believe that these bacteria were related to tourists? And gut microbes exist inside the body, how they invade the environment when visiting? Such putative explanation always lacks of enough direct evidence, please refer to more related references.

9. More latest references that focused on cultural heritage microbial communities and environmental parameters are recommended to add in the Introduction and Discussion part, e.g. the work of He et al., 2021, 2022 at Maijishan Grottoes, published on International Biodeterioration & Biodegradation.

Author Response

(The authors gave the same response as above.)
